# The Design of an Automatic Temperature Compensation System through Smart Heat Comparison/Judgment and Control for Stable Thermal Treatment in Hyperthermic Intraperitoneal Chemotherapy (HIPEC) Surgery

**DOI:** 10.3390/s23156722

**Published:** 2023-07-27

**Authors:** Kicheol Yoon, Sangyun Lee, Tae-Hyeon Lee, Kwang Gi Kim

**Affiliations:** 1Medical Devices R&D Center, Gachon University Gil Medical Center, 21, 774 Beon-gil, Namdong-daero, Namdong-gu, Incheon 21565, Republic of Korea; kcyoon98@gachon.ac.kr (K.Y.); l0421h@gmail.com (S.L.); 2Department of Biomedical Engineering, College of Medicine, Gachon University, 38–13, 3 Beon-gil, Dokjom-ro 3, Namdong-gu, Incheon 21565, Republic of Korea; 3Department of Electronic Engineering, Gyeonggi University of Science and Technology, Gyeonggigwagi-dearo 269, Siheung City 15073, Gyeonggi-do, Republic of Korea; thlee@gtec.ac.kr; 4Department of Biomedical Engineering, College of Health Science, Gachon University, 191 Hambak-moero, Yeonsu-gu, Incheon 21936, Republic of Korea; 5Department of Health Sciences and Technology, Gachon Advanced Institute for Health Sciences and Technology (GAIHST), Gachon University, 38–13, 3 Beon-gil, Dokjom-ro, Namdong-gu, Incheon 21565, Republic of Korea

**Keywords:** HIPEC surgery, high-temperature drug, cancer, LUT, temperature adjustment

## Abstract

After surgery for ovarian cancer or colorectal cancer, residual tumors are left around. A practical way to treat residual tumors is to destroy them with heat by injecting high-temperature drugs into the abdominal cavity. The injected medicinal substances are induced to flow out of the abdominal cavity; then, the spilled drug flows back into the abdominal cavity through feedback. During this process, the heat starts to decrease; thus, the treatment performance reduces. To overcome this problem, this study compares and assesses the temperature needed to maintain the heat for treatment and transmits a command signal to the heat exchanger through a look-up table (LUT). When the temperature decreases during the circulation of medications leaking out of the abdominal cavity, the LUT transmits a control signal (*T_p_*) to the heat exchanger, which increases or vice versa. However, if the temperature (*T_o_*) is within the treatment range, the LUT sends a *T_s_* signal to the heat exchanger. This principle generates a pulse signal for the temperature difference (*T_dif_*) in TC by comparing and determining the temperature (*T_o_*) of the substance flowing out of the abdominal cavity with the reference temperature (*T_ref_*) through the temperature comparator (TC). At this time, if the signal is 41 °C or less, the LUT generates (heats) a *T_p_* signal so that the temperature of the heat exchanger can be maintained in the range of 41 °C to 43 °C. If the *T_dif_* is 44 °C or higher, the LUT generates (cools) the T_a_ signal and maintains the temperature of the heat exchanger at 41–43 °C. If the *T_dif_* is maintained at 41–43 °C, the LUT generates a *T_x_* signal to stop the system performance. At this time, the TC operation performance and *T_dif_* generation process for comparing and determining the signal of *T_o_* and *T_ref_* for drugs leaking out of the abdominal cavity is very important. It was observed that the faster the response signal, the lower the comparison and judgment error was; therefore, the response signal was confirmed to be 0.209 μs. The proposed method can guarantee rapid/accurate/safe treatment and automatically induce temperature adjustment; thus, it could be applied to the field of surgery.

## 1. Introduction

As of 2017, there have been 232,255 cases of domestic cancer incidence, with an estimated cancer incidence rate of approximately 1,867,405 [1]. Therefore, approximately 3.6% of patients were diagnosed with cancer, which was the leading cause of death among men and women in 2018 [2]. After cancer surgery, a high probability of recurrence (within 5 years) at more than 50% exists due to the residual tumor tissue. 

Chemotherapy is administered concurrently to reduce the chance of recurrence. However, due to its side effects, patients experience difficulties during treatment. Thus, new treatment methods for eliminating the probability of recurrence within 5 years have been studied and are currently being used in many medical institutions [1,2,3,4,5,6]. Most of the thermal death methods for tumors include HIFU (High-Intensity Frequency Ultrasonic) treatment, laser treatment, and radiofrequency treatment [3,4,5,6]. Among them, the high-frequency treatment causes hyperthermia by an antenna that heats through radio and microwave radiation and damages cancer cells with heat.

In particular, HIFU can be applied to targeted therapy and can necrotize tumors using high heat [7]. This method can destroy cancer cells by generating immediate heat. However, it has the side effect of damaging the skin and its surrounding tissues [7]. Laser treatment causes a chemical reaction between light and oxygen to generate instant heat [8]. This method damages the micro-vessels around cancer cells to minimize patient pain and block the supply of nutrients to cancer tissue. These treatments cause side effects such as skin hypersensitivity, edema, pain, and pigmentation [9]. Laser treatment can also damage the skin [8,9].

Radiofrequency treatment is a method for irradiating cancer tissue with electromagnetic waves to generate heat, increase the metabolic rate in the body, and suppress the proliferation of cancer cells to destroy them [10]. Therefore, the radiofrequency treatment has no side effects such as nausea, vomiting, anorexia, weight loss, digestive disorders, and hair loss. Nevertheless, red spots may occur on the skin tissue where the electrodes are attached, and side effects such as slight burns, scars, inflammation, and lumps may occur in the fat layer [10,11].

In addition, HIFU therapy, laser therapy, radiofrequency therapy, and radiation therapy methods are used for the targeted treatment of tumors and are often used in combination with chemotherapy to increase the effectiveness of treatment. Chemotherapy is administered intravenously; however, various treatment systems that allow for the direct exposure of anticancer drugs are currently under investigation and development.

Among them, hyperthermic intraventricular chemotherapy (HIPEC) surgery is a system that uses high-temperature anticancer drugs and injects them into the abdominal cavity, and it is a treatment method that directly exposes anticancer drugs to residual tumors after surgical operation. Hyperthermia using high-temperature anticancer drugs has been shown to be highly effective against cancer cells when administered at temperatures ranging from 41 °C to 43 °C Celsius, whereas normal tissues can be damaged by temperatures above 46 °C Celsius [12,13,14]. This hyperthermia increases the sensitivity of cancer to chemotherapy by increasing the penetration of chemotherapy on the peritoneal surface and impairing DNA repair. In addition, hyperthermia has direct cytotoxic effects by inducing apoptosis, activating heat shock proteins that act as receptors for natural necrotic cells, inhibiting angiogenesis, and promoting protein denaturation [15]. The problem with these cancers is that the 5-year survival rate is difficult to expect, and the risk of recurrence is high; therefore, the purpose of HIPEC surgery is to completely remove the remaining cancer cells. The 5-year survival rate of gastric cancer patients undergoing HIPEC surgery increased by more than 61% compared to patients undergoing laparoscopic surgery [16,17,18,19]. Similarly, the 5-year survival rate of peritoneal cancer patients increased by over 48.5% compared to those undergoing laparoscopic surgery. Following colon cancer surgery, patients who received chemotherapy had a 41% recurrence rate at the 5-year survival mark due to peritoneal metastasis [20]. However, HIPEC surgery resulted in a 53% reduction in its recurrence rate at the 5-year survival rate.

However, if a 41 °C to 43 °C medication is injected into the abdominal cavity while using a hyperthermia system in the operating room, the intraperitoneal temperature is lowered to below 41 °C, and the therapeutic effect of hyperthermia cannot be enhanced [12].

In this study, when administering high-temperature medications in the abdominal cavity, the thermal compensation based on the temperature measurement could maintain the temperature of the drug at 41–43 °C by monitoring the temperature state of the drug and actively increasing the heat when the temperature decreased. Moreover, we developed the control method design. 

A control signal was generated to compensate for the temperature difference in the look-up table (LUT) module by comparing the temperature of the drug flowing out and the reference heat ranging from 41 °C to 43 °C. Hence, this study presents a method that compensates for heat according to the control signal.

## 2. Materials and Methods

### 2.1. Background of HIPEC Surgery

HIPEC treatments are categorized into open and closed techniques, as shown in Figure 1. The closed technique involves inserting and securing the inflow and outflow catheters into the abdominal cavity before injecting chemotherapy, then closing the abdominal wall and injecting the hot chemotherapy solution to allow for perfusion into the abdominal cavity. The main disadvantage of the closed technique is the uneven distribution of chemotherapeutic agents in the abdominal cavity, which results in fluid retention and the accumulation of toxic concentrations of drugs and heat [21].

In comparison, the HIPEC open method or “coliseum” technique involves injecting the hyperthermic chemotherapy solution into the abdominal cavity while the abdomen is open. The advantages of the open approach include the surgeon’s direct access to the abdominal cavity with an inflow catheter during the administration of the hyperthermic agent, which allows for the rapid and uniform control of the temperature and distribution of the drug in the abdominal cavity by manipulating fluids and the bowel. Care also needs to be taken to ensure that all peritoneal surfaces are uniformly exposed during treatment and to avoid dangerous temperatures or excessive exposure to normal tissue. The potential disadvantages of this procedure include the rapid loss of heat, which requires more effort to maintain ideal temperatures, and the potential exposure of the surgeon and operating staff to chemotherapeutic agents by direct contact and aerosolized particles [22].

For the use of these techniques, a HIPEC system consists of a fluid box, a heat exchanger, and a filter, as shown in Figure 2. It is further composed of inlet and outlet catheters that are connected to the system [15,21,22,23].

**Figure 1 sensors-23-06722-f001:**
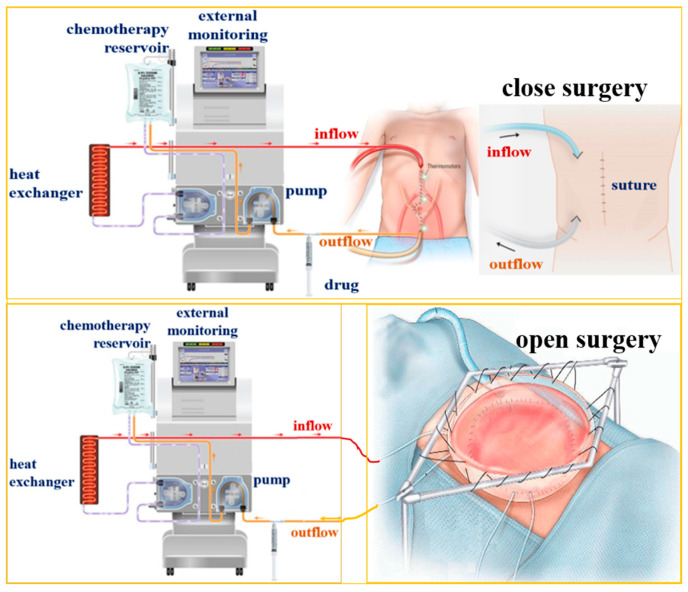
Concept of HIPEC surgery [24].

The heat exchanger operates to heat the drug to a high temperature. The medicinal substances flow into the heat exchanger and are heated up to 48 °C. The heated drug is introduced into the abdomen through an inflow catheter, and the abdominal temperature remains between 41 °C and 43 °C for 120 min, damaging cancer cells remaining in the abdomen [15,21,24].

The anti-cancer drug that flows into the abdomen continues to flow into the fluid bow through the filter and outflow catheter. Additionally, the medicinal substances that flow in are repeatedly circulated through the inflow and outflow of the abdomen through the heat exchanger and inflow catheter. At this point, the filter operates to filter out the mixed residues from the abdomen in the drug leaked from the outflow catheter. When performing thermal treatment using the HIPEC system, the temperature of the abdomen should be maintained between 41 °C and 43 °C [15,21,22,23,24]. However, if the medication passes through the line in the process of repeating the inflow and outflow of the medicinal substances or if thermal treatment is performed through open surgery, the temperature of the drug in the abdominal cavity can drop to 36 °C. If the drug temperature drops to below 41 °C, the tumor does not undergo necrosis and rather results in poor treatment. Therefore, for efficient thermal treatment, a functional device for maintaining the temperature of the interior of the abdominal cavity at 41 °C to 43 °C is required. In the operating room, as shown in Figure 3, if the temperature in the abdominal cavity actually decreases below 41 °C, heat is manually controlled [21,22,23,24].

If the temperature increases by more than 43 °C, the treatment process can be stopped until the temperature decreases [15,21,22,23,24]. The abovementioned actions are cumbersome and lead to time and human force consumption. Thus, there is a need to examine an automatic control method to adjust the proper temperature by real-time temperature sensing and compensation, which is capable of maintaining the treatment temperature.

### 2.2. Design of HIPEC System for Temperature Comparison and Control

As shown in Figure 4, the reference heat generator (HG) that is intended to maintain the intraperitoneal medicinal substances temperature generates heat (TR) within a range of 41–43 °C. This heat (TR) is converted into an analog signal through a temperature-to-voltage converter (TVC).

The analog signal provided by the TVC is converted into a pulse train (*T_ref_*) by an analog-to-digital converter (ADC). The heat (*T_o_*) leaking from the abdominal cavity is detected by the temperature sensor. The detected temperature *T_o_* is converted into an analog signal through the TVC [24]. This analog signal is converted into a pulse signal via ADC for feedback. The *T*_0_ signal is converted into a pulse signal, which is input into the TC and generates a *T_dif_* signal, as shown in Equation (2) through the comparison and judgment with *T_ref_* as shown in Equation (1), where h and *χ* represent Joule’s heat and thermal conductivities with respect to heat *T_o_*, respectively.
(1)$$T_{o}\left(t\right)=T_{dif}\sin\left[T_{ref}\left(t\right)+h\left(t\right)\right]=\chi T_{dif}\left(t\right)\sin\chi T_{ref}h\left(t\right)+\frac{1}{\sqrt{\frac{1+tan\left(\frac{h_{dif}\left(t\right)}{2}\right)}{1-tan\left(\frac{h_{ref}\left(t\right)}{2}\right)}}}\left[ln\left(1-\frac{\left[1+tan\left(\frac{h_{dif}\left(t\right)}{2}\right)\right]}{4\pi \left[1-tan\left(\frac{\chi T_{ref}h\left(t\right)}{2}\right)\right]}\right)\right]-\left[2ln\frac{T_{ref}}{T_{ref}-T_{dif}}\right]$$
(2)$$T_{dif}\left(t\right)=\chi T_{ref}h\left(t\right)-\chi T_{o}\left(t\right)=\chi T_{ref}h\left(t\right)-2\pi \chi T_{o}\int_{-\infty }^{\infty }T_{o}dt=\int_{0}^{t}T_{o}\left(h\left(t\right)-1\right)dt=2\pi \chi T_{ref}h\int_{0}^{t}T_{o}dt$$

In Equation (1), changes occur over time in a limited temperature range by the function of *T_ref_*(*t*). However, *T_ref_* changes within a temperature range of 41–43 °C. In this case, the temperature range must have excellent thermal conductivity (*χ*) in order to accurately maintain 41–43 °C. That is, h is defined as a Joule heat function for fixing a temperature range of 41–43 °C. *T_ref_* with excellent thermal conductivity (*χ*) increases its temperature by tan over time, and the temperature change in *T_ref_* increases by *tan*(*t*) and is maintained within a certain range through saturation. This allows changes to be made, allowing the temperature to rise, fall, and be maintained over time. Therefore, the sum of *T_ref_* and *T_dif_* (*T_ref_* + *T_dif_*) can observe the *T_o_* temporal state of change. T_dif_ arises from the difference between *T_ref_* and *T_o_*. At this time, if the temperature (*T_ref_*) that changes over time and the temperature (*T_o_*) that changes over time in the *h*(*t*) function range with excellent thermal conductivity (*χ*), they generate a temperature difference in TC, *T_dif_*(*t*) that changes from 0 to *t*, as shown in Equation (2).

The signal (*T_dif_*) from the TC is input to a lookup table (LUT). As shown in Table 1 and Figure 5, the LUT generates a control command signal (*ord*) through internal operations. This command signal (*ord*) is delivered to a heat exchanger, which heats up (*T_p_*), reduces heat (*T_a_*), stops operating (*T_x_*), or maintains its current state (*T_s_*). If *T_o_* is below 41 °C, *T_dif_* for this temperature (41 °C) passes to the LUT, which generates signal *T_p_* to command the heat exchanger. The heat exchanger compensates the heat to reach the therapeutic temperature range (41–43 °C). If *T_o_* is within this range, the LUT generates signal *T_s_* (hold state), and the operation of the heat exchanger does not change. If *T_o_* is above 44 °C, the LUT generates signal *T_a_*, and the heat exchanger reduces the temperature. If *T_o_* exceeds 46 °C, the LUT generates signal *T_x_*, and the heat exchanger stops operating (i.e., stop heating and wait).

As shown in Figure 5, *T_ref_* and *T_dif_* repeat 0 and 1, respectively, and the signal is input to the TC. *T_dif_* is divided into state maintenance, temperature reduction (0), temperature increase (1), and operation interruption (x).

Figure 6 shows the circuit for the comparator and LUT, which includes a non-exclusive OR gate.

When the medication flows out of the abdominal cavity, the temperature (*T_o_*) flowing out is compared with the temperature of the *T_ref_* through the TC to the output *T_dif_*. At this time, TC is in charge of the performance that generates *T_dif_* through the comparison and judgment of *T_ref_* and *T_o_*. *T_dif_* is then input to LUT to generate the command signals (*ord*) of *T_a_*, *T_p_*, *T_s_*, and T_x_. The signals (*T_p_*, *T_s_*, *T_a_*, and *T_x_*) command the heat exchanger for temperature control. The proposed system is a chemical surgical treatment that removes cancer by injecting high-temperature medicinal substances into the abdominal cavity. The drug maintains the drug’s temperature in the range of 41–43 °C through intraperitoneal injection and drainage. The drug circulates through a heat exchanger. These performances are automatically and repeatedly controlled until the treatment is complete.

## 3. Experimental Results

For the performance test of the substance temperature maintenance, we conducted an experiment that comprised the proposed module, beaker, thermal imaging camera, and temperature measuring device, as shown in Figure 7. This study was not a clinical trial but an experiment to test the performance of the module and evaluate whether the heat temperature could be accurately controlled. Therefore, we aimed to obtain results on the accuracy, reproducibility, and performance reliability of temperature control. Additionally, the performance test evaluation to minimize the response error (delay) for control was reviewed.

The peritoneal cavity could be substituted with a beaker (100 mL) when filled with normal saline (40 cc). The temperature range of normal saline is 41–43 °C; if this was below 41 °C or above 43 °C, the TC and LUT were operated and controlled to match this range. Thus, it was simulated similarly to the HIPEC surgical procedure.

The designed temperature control module and the beaker were connected with a catheter, as shown in Figure 7, and the catheter functioned as the inflow and outflow of the beaker. An external monitoring system measured the temperature conditions and thermal compensation adjustments.

The system determined the pulse signals according to the current drug temperature to reduce, increase, or maintain the temperature. In temperature tests, the pulse signal range of *T_ref_* was stably maintained in the range of 41–43 °C, as shown in Figure 8. If the temperature of the signal *T_o_* was reduced to 41 °C or less, the pulse signal of the different output from the TC was set to transmit to the LUT. Thus, the LUT generated a command signal (*ord*) for the *T_p_* signal and took measures to heat it in the heat exchanger.

The proposed module configured *T_ref_* and *T_o_* signals before their input to TC. The TC-*T_dif_* output worked to output a pulse signal for the drug temperature state. As shown in Figure 8 and Figure 9, when the pulse signal (*T_dif_* = *T_ref_* = *T_o_*) corresponding to the range of *T_dif_* from 41 °C to 43 °C was output, the output pulse signal was 1. At this time, as shown in Figure 8, the signal of 1 was in the form of a pulse corresponding to 4 to 5 μs. If the *T_dif_* (*T_dif_* = *T_o_*/*T_ref_*) signal output from the TC was below 41 °C, the pulse signal of *T_dif_* corresponded to 1, and this pulse signal corresponded to a period of 2–4 μs. If the signal output from the TC *T_dif_* (*T_dif_* = *T_o_*/*T_ref_*) corresponded to 44 °C or higher, the pulse was 0. The period of this signal is 1–2 μs. The signal output by *T_dif_* (*T_dif_* = *T_o_*/*T_ref_*) in TC was a pulse signal of 1 corresponding to 46 °C or higher, and the period of this pulse signal corresponded to 7–8 μs. Therefore, the pulse signal of 1 (4–5 μs) generated from *T_dif_* was delivered to the LUT, and the LUT generated a *T_s_* signal (41–43 °C); therefore, the substance was injected into the abdominal cavity without being transmitted to the heat exchanger.

The pulse signal corresponding to 1 (2–4 μs) and generated from *T_dif_* was delivered to the LUT. The LUT generated a *T_p_* signal (<41 °C). The *T_p_* signal provided the heat exchanger with a command signal for heating to maintain a temperature range of 41 °C to 43 °C. The pulse signal of 0 (1–2 μs) generated from *T_dif_* was delivered to the LUT. The LUT generates a *T_a_* signal (>44 °C). The *T_a_* signal delivers a command signal for cooling to maintain a temperature from 41 °C to 43 °C in the heat exchanger.

The pulse signal of 1 (7–8 μs) generated from *T_dif_* was delivered to the LUT, and the LUT generated a *T_x_* signal (>46 °C), which transmitted the *T_x_* signal to the heat exchanger. Therefore, the system operation was stopped until the intraperitoneal injection temperature range changed from 41 °C to 43 °C. Therefore, the command signal data transmitted from the LUT to the heat exchanger were *T_p_*, *T_s_*, *T_a_*, and *T_x_*.

More specifically, if the logical function of the D-flip flop was considered in the experiment, the control process of TC and data learning of LUT was added to the heat exchanger using a binary signal and based on a pulse waveform. Therefore, the temperature of the drug in the heat exchanger could be controlled, as shown in Figure 9 and Figure 10. If *T_ref_* always maintains a constant value in the range of 41—43 °C (*T_ref_* = 0), the range of the temperature of the medicinal substance (*T_dif_*) flowing out of the beaker (peritoneal cavity) is 41—43 °C (T_o_ = 0), as shown in Figure 10. At this point, the temperature difference (*T_dif_* = *T_o_* = *T_ref_*) generated in the comparator could provide a value of 1. Thus, the temperature of the substance in the heat exchanger remained unchanged, and the temperature range of T_dif_ was between 41 °C and 43 °C. In the experimental process of the temperature difference generation for the TC output, *T_ref_* and *T_o_* were assumed to be 38 °C (*T_o_* = 0) and 41 °C (*T_ref_* = 1), respectively, and then these two pulse signals were set to be input to TC. The output (*T_dif_*) of the TC generated a pulse signal (*T_dif_* = random signal) in which a temperature difference of 3 °C occurred. When these signals were input to the LUT, the LUT recognized a temperature difference of 3 °C. The LUT, which recognized a temperature difference of 3 °C, generated a command signal (*ord*) of *T_p_* in the output. This command signal (*ord*) could be input to the heat exchanger, which heat unless at least 3 °C to reach 41 °C.

It can be noted that *T_a_*, *T_p_*, *T_s_*, and *T_x_* were accurately output when the pulse waveform signals corresponding to 41 and 43 °C were periodically generated in *T_ref_*. When *T_ref_* and *T_o_* were compared in the D-flip flop, the signal generated from *T_dif_* could minimize the delay signal (*T*_1_ and *T*_2_), as shown in Figure 11. Therefore, the signal (*T_(t)_*) generated from *T_dif_* could generate an accurate signal without an error, as shown in Equation (3) [25].
(3)$$T\left(t\right)=\frac{\left(T_{2}+T_{1}\right)}{T_{r}}-\left(1-e^{-dt}\right)$$

Here, *T_(t)_* represents the period of the delay signal for the difference between *T*_1_ and *T*_2_, and *T_r_* represents the theoretical minimum delay time. As shown in Figure 11, *T*_1_ and *T*_2_ generated 209 ns each, and *T_r_* generated 4.18 ps. Moreover, the error for the delay time of the module was within 1.8%. Therefore, the accuracy, reliability, and reproducibility of the module’s operation had a high level of 98.2%, respectively.

The control performance (inflow, outflow, and feedback) was tested by connecting the designed module to a beaker. Figure 12a presents an image that was obtained using a thermal imaging camera for the thermal temperature state of normal saline inside the beaker. We also evaluated the correctness of signals *T_a_*, *T_p_*, *T_s_*, and *T_x_* obtained from the LUT through thermal imaging, confirming that the temperature of the drug was maintained in the therapeutic range by the system. Additionally, the thermal temperature distribution signal for the thermal temperature control performance according to the pulse waveform is shown in Figure 12b,c. The results show that, over time, the heat temperature control of the heat exchanger was well-adjusted so that it could be properly maintained.

When the *T_dif_* signal was generated from the TC, the LUT provided *T_a_*, *T_p_*, *T_s_*, and *T_x_* signals to control the temperature of the heat exchanger (shown in Figure 13) and supply constant heat from the heat exchanger during time changes. By performing the interlocking function of TC and LUT, the heat distribution graph showed that the control system maintained its temperature within the desired range. Therefore, this result confirmed that the operation was suitable for HIPEC’s surgical treatment.

Finally, Figure 14 shows the connected circuits that were configured to verify the proposed design method for the three-dimensional (3D) printer, the PCB (printed circuit board) board, and the motor, and evaluates whether the temperature was controlled using a temperature sensor, monitor device, and thermal imaging camera. For the operation of the evaluation, the supplementary version of this discussion can be referred to, and when the temperature fell below 41 °C (*T_p_*) in the equipment, the control system automatically raised the heat to reach the range of 41–43 °C (*T_s_*). If the heat reached between 44 °C and 46 °C (*T_a_*), the control system automatically stopped the operation (*T_x_*) and then adjusted to stay at 41 °C to 43 °C (*T_s_*). Therefore, the measurement results were consistent with the experimental (see Figure 13) and simulation results (see Figure 7, Figure 8, Figure 9, Figure 10, Figure 11, Figure 12 and Figure 13). When operating for 120 min in the control system so that the temperature could reach a stable temperature (*T_s_*) at a range between 41 °C and 43 °C [15,21,22,23,24] and a heat temperature of 42 °C after 5 min in the temperature meter. This was measured, and after 50 min, a 43 °C measurement result was obtained from the thermal imaging camera. When considering the operation time of 120 min [15,21,22,23,24], the measurement results were analyzed to be controllable in a sufficiently stable state.

## 4. Discussion

In current surgical procedures, the HIPEC treatment method manually controls the temperature with an assistant. If the heat rises, it starts to lower the heat; if the heat decreases, the assistant manually adjusts to increase the heat. In addition, if the fever is constant, the system is on standby; if the fever starts to decrease or increase, the treatment has to stop until the proper temperature is maintained. Therefore, the treatment process is cumbersome and time-consuming. However, this study has the advantage of automatically adjusting the heat through temperature monitoring to solve the complicated process. When comparing and judging temperatures and transmitting the provided data to the LUT to generate a command signal, the reaction speed was the crucial parameter. As the reaction speed increased, the operational error decreased. Moreover, the reaction speed is an essential factor because it is related to the accuracy and reliability of the system. In this regard, the proposed system has a high response speed and high reliability. The designed system can also be evaluated by experimenting with animals. However, our experiment of the designed system focused on maintaining the heat in the heat exchanger (41–43 °C) and evaluating the accuracy and reliability of the compensation function through comparison, judgment, LUT data learning, and the generation of a light signal by sensing heat. Note that evaluating the accuracy and reliability of the compensation function is paramount. More specifically, when the temperature is too low or too high, the pulse signal conversion timing for the comparison and discrimination to reach the treatment temperature range (41–43 °C) through the TC control is considered to be very important. The reason for this is that the comparison and judgment speed for the control can be related to the performance of TC’s control speed. Therefore, the fast operation of the heat exchanger by sending a command signal (*ord*) to the LUT through a quick comparison and judgment of the TC is related to the operation time. The operation time was 90–120 min in total, and during that time, it was very important to reduce the delay time of the signal response characteristics to increase the patient’s treatment performance through rapid control and circulation. Thus, the experiments and functional tests used a beaker and a thermal imaging camera, which were considered appropriate methods to evaluate the proposed system. Nevertheless, animal experiments are necessary for commercialization in the future. As shown in Table 2, the proposed method is estimated to be 51 times faster than previously studied methods, considering the system response speed. The data for the comparison and judgment obtained in this study included binary signals detecting heat. The cases studied in Table 2 are modules designed for error correction through comparison, judgment, and feedback on binary signals rather than a system for thermal treatment. Consequently, the important part of these modules was the response speed of the signal. Hence, improving the response speed is significant for comparison and judgment. In Table 2, the response speed of the proposed system outperforms that of the response speed in the existing literature. As shown in Figure 11, *T*_1_ and *T*_2_ generated 209 ns each, and T_r_ generated 4.18 ps. The reason for this is that *T*_1_ and *T*_2_ should be theoretically reached within 1 μs (*T_r_* = 1 μs), but according to cases announced in 2021, their delay time was within 225 ns to 73 μs. Thus, the smaller the delay time, the better the module’s performance, allowing it to become superior [26,27,28,29,30,31]. In particular, the error for the delay time of the module was 1.8% compared to the delay time in [26], which could reach the 98.2% level for the accuracy, reliability, and reproducibility of the module [32,33].

## 5. Conclusions

In this paper, a control system was proposed to maintain and compensate the treatment temperature in HIPEC surgery using high-temperature drugs for ovarian cancer surgery.

The small tumor tissue remaining after ovarian cancer surgery is difficult to treat surgically. Therefore, the method proposed in this study can increase the therapeutic effect of hyperthermia chemotherapy by injecting and circulating the drug in the abdominal cavity and keeping the treatment temperature constant. In addition, by preventing the temperature from rising above a certain point, it can reduce the side effects caused by damage to normal tissues.

During the treatment, the temperature of the medicinal substances should always be constant; however, the fever may be lowered because the drug circulates through and outside the body. Conversely, heat may increase. Therefore, if the heat is manually controlled by monitoring the temperature from the outside, this process has the disadvantage of interfering with the treatment. However, the advantage of our study is that the heat can be automatically reduced through external monitoring. Alternatively, the heat can be automatically increased. Additionally, the proposed system can retain heat and has an automatic control function. The method for maintaining/compensating for the temperature through heat comparison/determination, data collection/learning of the LUT, and the control signal output is considered a crucial and groundbreaking technology. Further, the response time for the proper control and comparison, and judgment of heat was substantially short in the feedback process because the time required for comparison and judgment was minimal. Moreover, the system had a low margin of error (within 1.8%) and high accuracy (98.2%). The application of this technology can eliminate errors in manual methods controlled by humans and guarantee accurate, safe treatment and rapid surgery. In conclusion, the proposed system has good application prospects and could be applied to the field of surgery, obstetrics, and gynecology.

## Figures and Tables

**Figure 2 sensors-23-06722-f002:**
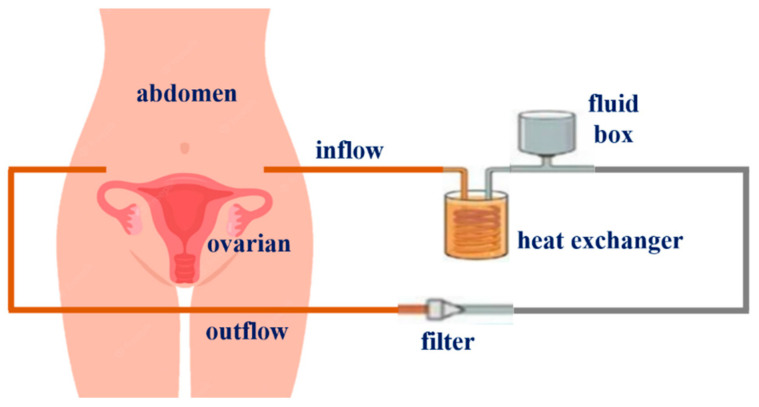
Principle of drug delivery during HIPEC surgery.

**Figure 3 sensors-23-06722-f003:**
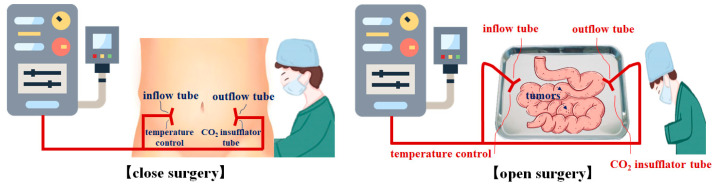
HIPEC surgery (operating room) [21].

**Figure 4 sensors-23-06722-f004:**
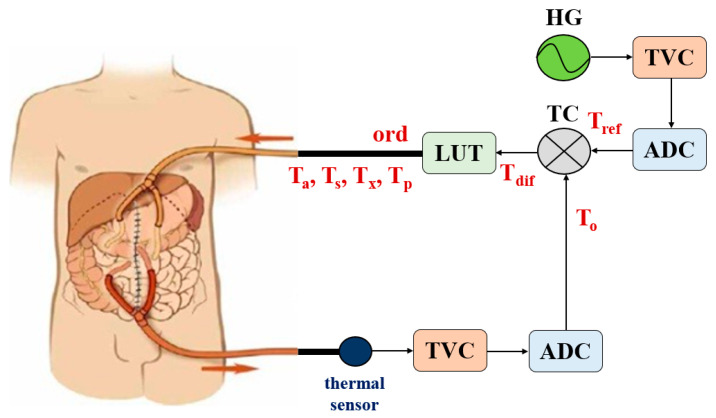
Block diagram of proposed system.

**Figure 5 sensors-23-06722-f005:**
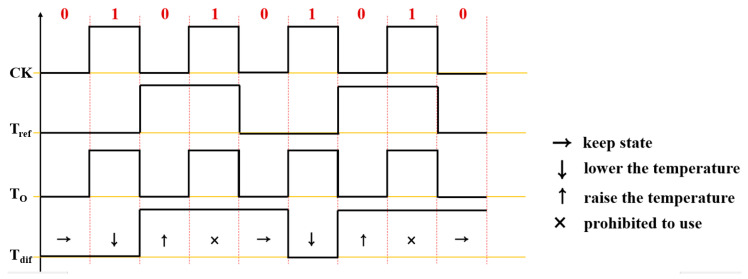
Logic signals for temperature comparison and decision by the proposed system.

**Figure 6 sensors-23-06722-f006:**
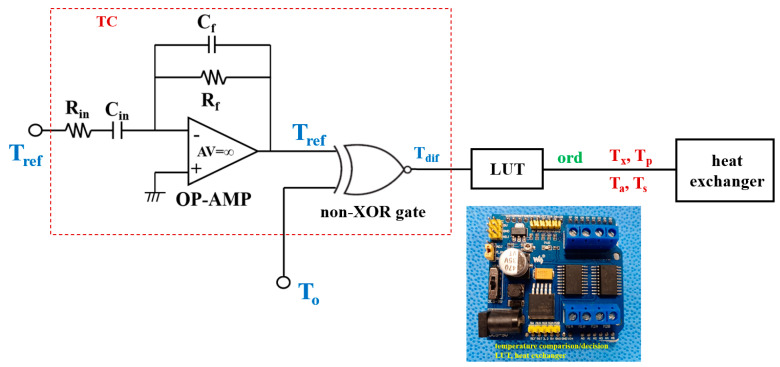
Equivalent circuit of proposed system for temperature comparison, control and the implemented circuit.

**Figure 7 sensors-23-06722-f007:**
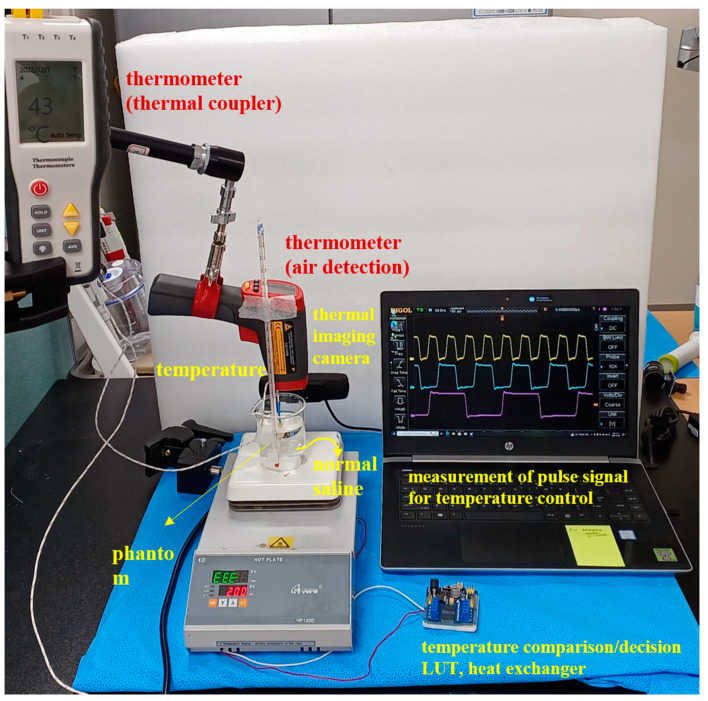
Experimental environment and configuration system.

**Figure 8 sensors-23-06722-f008:**
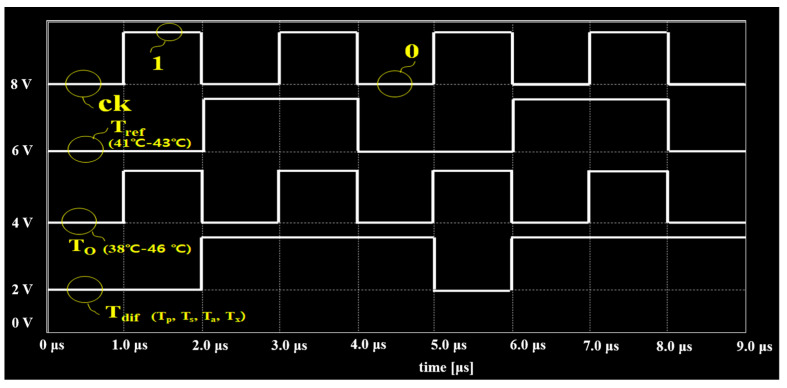
Output of pulse logic signal for the comparison and control of temperature based on TC.

**Figure 9 sensors-23-06722-f009:**
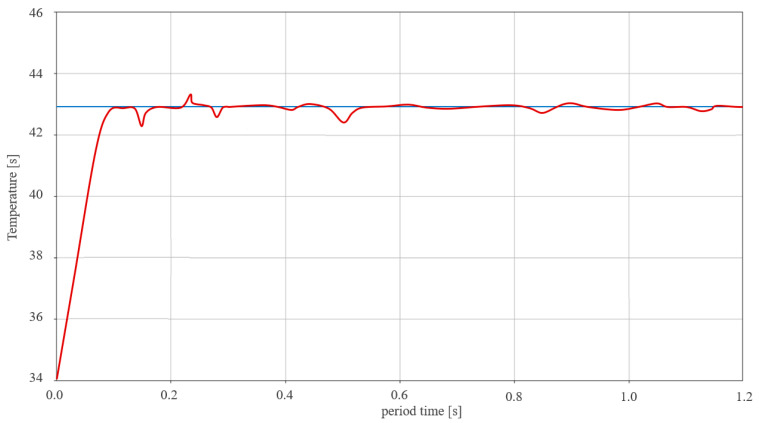
Output temperature obtained from the comparison and control based on TC.

**Figure 10 sensors-23-06722-f010:**
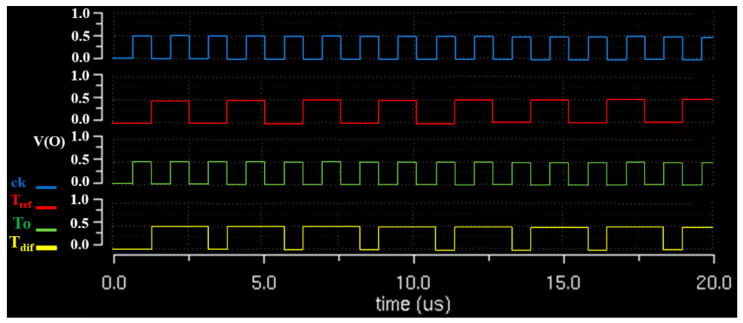
Measurement results for the pulse signal corresponding to temperature.

**Figure 11 sensors-23-06722-f011:**
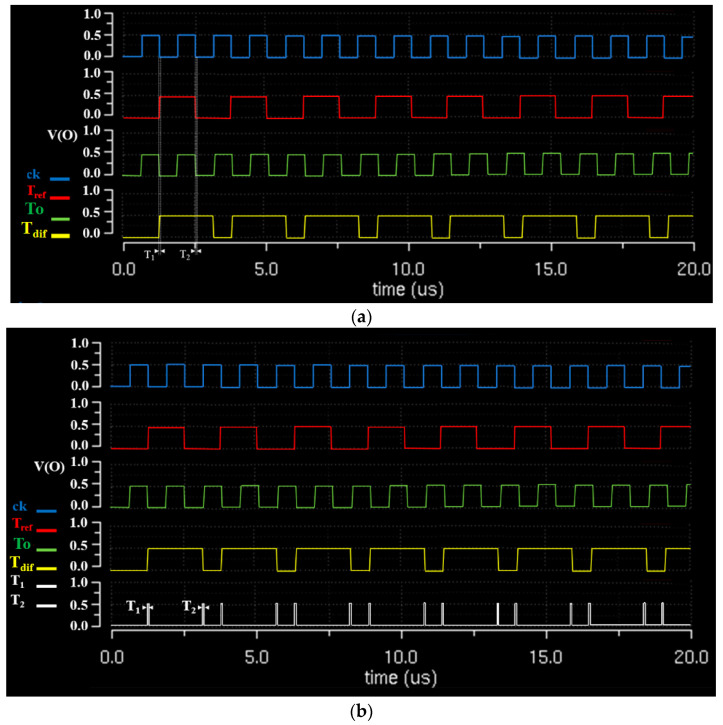
Measurement results of the pulse signal with response delay in parallel with the temperature comparison and decision (T_dif_). (**a**) Response delay; (**b**) Sampling of response delay.

**Figure 12 sensors-23-06722-f012:**
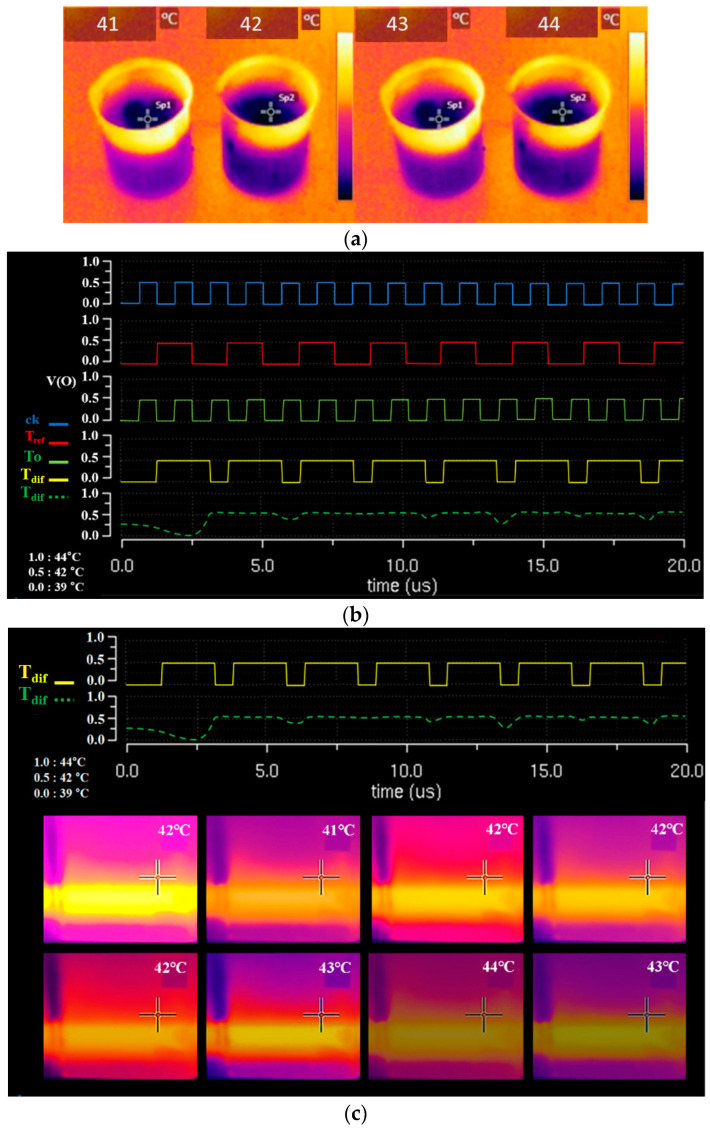
Comparison for measured output pulse signal, thermal condition graph, and thermal imaging camera. (**a**) Shooting of thermal imaging (high temperature of normal saline); (**b**) Pulse signal and thermal condition graph; (**c**) Pulse signal and thermal imaging camera.

**Figure 13 sensors-23-06722-f013:**
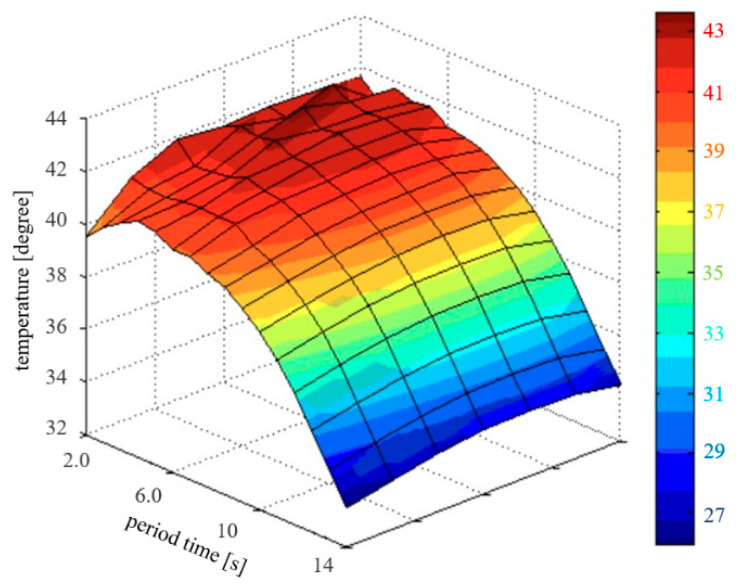
Three-dimensional (3D) graph based on experimental results verifying that the temperature was maintained.

**Figure 14 sensors-23-06722-f014:**
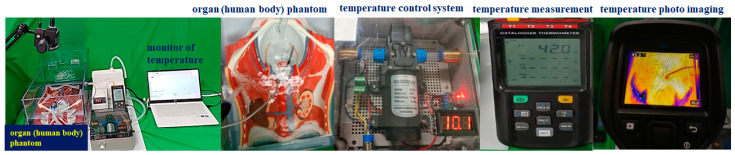
Proposed system fabrication and temperature test.

**Table 1 sensors-23-06722-t001:** Design of the look-up table (LUT).

Temperature Range	Signal to LUT			Status of Temperature	
	*T_ref_*	*T_o_*	*T_dif_*	*ord*	TC
40 °C	1	0	–	up	T_p_
41–43 °C	0	0	1	hold	T_s_
44 °C	0	1	0	down	T_a_
>46 °C	1	1	x	stop operation	T_x_

**Table 2 sensors-23-06722-t002:** Comparison of response time difference for proposed and existing systems.

Reference	Response Speed [μs]	Response Speed Difference [μs]	Performance Difference [times]
[26]	0.225	0.016	1.08
[27]	73.1	72.891	349
[28]	12.8	12.591	60.8
[29]	5.00	4.791	23.0
[30]	30.0	29.791	143
[31]	8.00	7.791	37.3
this work	0.209	—	—

## Data Availability

The data presented in this study are available upon request from the corresponding author. The data are not publicly available because of privacy and ethical restrictions.

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
