# Peer review of "The Design of an Automatic Temperature Compensation System through Smart Heat Comparison/Judgment and Control for Stable Thermal Treatment in Hyperthermic Intraperitoneal Chemotherapy (HIPEC) Surgery"

_sensors, 2023, doi:10.3390/s23156722_

Round 1

Reviewer 1 Report

The authors raise a very interesting and important topic. However, I have a few reservations about the manuscript:

1. In the introduction, the authors provide statistical data on cancer incidence. I propose to specify these data to cancers that are treated with HIPEC

2. The text repeatedly mentions "tumor kill etc." I propose to change these statements to more professional and substantive vocabulary

3. The text repeatedly mentions "anti-tumor drugs". I suggest changing these statements to more professional and substantive vocabulary, e.g. "chemotherapeutics"

3. I suggest expanding the introduction more to provide a better background for the reader

In conclusion, the work presents a very interesting topic. However, significant lexical and grammatical improvement of the manuscript is required

Significant improvement in vocabulary and grammar is required

Author Response

Comments 1

Comments 0 :

The authors raise a very interesting and important topic. However, I have a few reservations about the manuscript:

Answer 0 :

Thank you very much for your interest in my thesis. We will try our best to respond to your comments. The author also read it in its entirety and made his own corrections. Note the gray highlights.

Comments 1 :

In the introduction, the authors provide statistical data on cancer incidence. I propose to specify these data to cancers that are treated with HIPEC

Answer 1 :

Added a sentence about HIPEC surgery survival statistics. Please refer to the lines (yellow) in introduction 90-99.

Comments 2 :

The text repeatedly mentions "tumor kill etc." I propose to change these statements to more professional and substantive vocabulary

Answer 2 :

It has been checked in its entirety and corrected in professional terms. For example necrosis, thermal death, thermal damage.

Comments 3 :

 The text repeatedly mentions "anti-tumor drugs". I suggest changing these statements to more professional and substantive vocabulary, e.g. "chemotherapeutics"

Answer 3 :

I made an overall correction. Redundant words have been reduced. And the word has been used in a number of ways, appropriately.

Examples: drug, medicinal substances, medication, and substance.

Comments 4 :

 I suggest expanding the introduction more to provide a better background for the reader

Answer 4 :

The introductory text has been expanded. I added lines 90-99 (yellow) in the introduction because the content of the hipec operation seems to be lacking.

Comments 5 :

In conclusion, the work presents a very interesting topic. However, significant lexical and grammatical improvement of the manuscript is required

Answer 5 :

Although the overall correction has been made, the English grammar is currently being corrected by an expert. We will do our best to correct the English grammar until it is accepted. First, I responded to your comments. Please review.

Comments (Comments on the Quality of English Language) :

Significant improvement in vocabulary and grammar is required.

Answer (Comments on the Quality of English Language) :

I received English correction through an expert, but I am currently having my English grammar corrected by requesting it again from an expert. We will do our best to correct the English grammar until it is accepted. First, I responded to your comments. Please review.

Reviewer 2 Report

The authors present a temperature steering system for HIPEC. The goal of this system is the stabilization of the temperature in the therapeutic range of 41-43°C.

The main critical point of the article is the difficulty of understanding the details. Thus, new variables appear without any definition or connection to other variables (e.g. TD in table 1, CK in figure 5 and Tin in figure 8). Also the explanations of the figures (e.g. 5 and 8) should be improved. Generally, better explanation is necessary for a better understanding of the article.

Another aspect is that there are additional reasons for uncertainties of temperature steering in HIPEC. E.g. the inhomogeneous temperature distribution of the drug in the body caused by different flux in different regions of the body. Therefore, the discussion of the “respond delay” is less important for the application in HIPEC.

Some further remarks:

-          line 2: I think incidence is better than incidence rate,

-          line 55: hyperthermia by antenna that heats by radio- and microwave radiation is missing,

-          line 75: radiation therapy generates no heat, it damages the cells by damaging the DNA directly,

-          there is also circulation in the closed system (see figure 1), in the text no circulation seem s to be a difference to the open systems,

-          are the equations (1)-(3) necessary? They are more confusing than helping,

-          line 156 is the same as line 159,

-          line 187: is T =T0?

-          line 222: figure 7 instead of figure 8.

Author Response

Comments 2

Comments 0 :

The authors present a temperature steering system for HIPEC. The goal of this system is the stabilization of the temperature in the therapeutic range of 41-43°C.

Answer 0 :

Thank you very much for your interest in my thesis. We will try our best to respond to your comments. The author also read it in its entirety and made his own corrections. Note the gray highlights.

Comments 1 :

The main critical point of the article is the difficulty of understanding the details. Thus, new variables appear without any definition or connection to other variables (e.g. TD in table 1, CK in figure 5 and Tin in figure 8). Also the explanations of the figures (e.g. 5 and 8) should be improved. Generally, better explanation is necessary for a better understanding of the article.

Answer 1 :

Figure 5 and Figure 8 have been modified to logically match. And overall, I made modifications to fit the content logically. Note the gray highlights. Thank you for your detailed review.

Comments 2 :

Another aspect is that there are additional reasons for uncertainties of temperature steering in HIPEC. E.g. the inhomogeneous temperature distribution of the drug in the body caused by different flux in different regions of the body. Therefore, the discussion of the “respond delay” is less important for the application in HIPEC.

Answer 2 :

Discussion

When the temperature is low or too high, the pulse signal conversion timing for comparison and discrimination to reach the treatment temperature range (41-43℃) through TC control is considered to be very important. The reason for this is that the speed of comparison and judgment for control is related to the performance of the TC control speed, so the fast operation of the heat exchanger by sending a command signal (ord) to the LUT through the quick comparison and judgment of the TC is also related to the operation time. there is. The operation time is 90-120 minutes in total, and during that time, it is very important to reduce the delay time of the signal response characteristics to increase the patient's treatment performance through rapid control and circulation. Therefore, I reflected the sentence in lines 375-384 (green) of the discussion session.

Some further remarks:

Comments 3 :

 line 2: I think incidence is better than incidence rate,

Answer 3 :

thanks for your advice As per your comment, I have made corrections to improve the syntax better. Please refer to line 46 (yellow) in the introduction.

Comments 4 :

 line 55: hyperthermia by antenna that heats by radio- and microwave radiation is missing,

Answer 4 :

As per your comment, I inserted a sentence in hyperthermia by antenna that heats by radio- and microwave radiation. Please refer to the lines 56-58 (light blue). thank you for the condition Also can you recommend a reference for my thesis? I need to do a citation for a sentence about your opinion.

Comments 5 :

line 75: radiation therapy generates no heat, it damages the cells by damaging the DNA directly,

Answer 5 :

As you said, radiation treatment directly damages DNA. Radiation therapy was deleted because it did not appear to be appropriate for my thesis. Please refer to lines 54-56 (green) in the introduction. thanks for your advice

Comments 6 :

there is also circulation in the closed system (see figure 1), in the text no circulation seem s to be a difference to the open systems,

Answer 6 :

Thank you for checking the corrections. Figure 1 has been modified.

Comments 7 :

 are the equations (1)-(3) necessary? They are more confusing than helping,

Answer 7 :

Eqs (1)-(3) have been reduced to Eqs (1)-(2). And the content has been revised throughout. Also, we added more details about Eqs (1)-(2). When I read the content again, it seemed that it was expressed without the need for an expression. Expressions are needed to help explain system behavior and are judged to be necessary to help explain and understand the comparison and judgment of TCs and the expression of definitions through circulation. Equations (1)-(2) describe the process of generating Tdif (temperature difference) after determining by comparing the signal (To) of the temperature of the drug flowing out through the drainage for drug injection and the signal of the reference temperature. Needed to help explain. Therefore, depending on the Tdif signal, whether or not the LUT command signal is generated is determined to be necessary. See lines 166-188 (light blue) for details on equations (1)-(2).

Comments 8 :

 line 156 is the same as line 159,

Answer 8 :

I didn't understand the intent of the question. Could you please elaborate again? If you let us know again, we will do our best to reflect it. Thank you.

Comments 9 :

  line 187: is T =T0?

Answer 9 :

T is Tdif. Corrected on line 203 (yellow). Thanks for the check and advice.

Comments 10 :

line 222: figure 7 instead of figure 8.

Answer 10 :

Figure 8 has been modified to Figure 7. Please refer to line 240 (pink). Thanks for checking.

Round 2

Reviewer 1 Report

The authors made all the changes mentioned in the manuscript. I believe the paper is suitable for publication

Author Response

Comments 1

Comments :

The authors made all the changes mentioned in the manuscript. I believe the paper is suitable for publication:

Answer :

Thank you very much for carefully reviewing and evaluating my paper.

Reviewer 2 Report

Caused by the addition of new text in the second version of the article one problem is now enhanced that should be considered.

In the introduction and in other parts of the article the authors should clarify and improve the explanation of the effect of hyperthermia in tumor treatment.

It is possible to “destroy” the tumor cells by heat, but for this, much higher temperatures than 41°-43° are necessary and it is usually called “thermoablation”. This could be done by HIFU, laser and RFA.

In the temperature range of 40°-43° - called hyperthermia – the effect is not directly the cell destroying but the amplifying of the direct cytotoxic effect of chemotherapy and/or radiation therapy. This is also in HIPEC the effect and the goal. Therefore, formulations as “damage tumor cells by heat” or “thermal necrosis” are misleading in this temperature range and should be corrected.

Some further remarks:

·         line 23: substances are

·         line 35: I think: …  If Tdif is 44°C or higher  

·         line 75: radiation therapy does not cause heat necrosis but DNA damage that results in necrosis

·         line 102: “reducing the effect of the therapeutic agent …” instead of “ making it impossible …” and no thermal (see above)!

·         lines 115-126: In the text the main difference between close and open HIPEC seems to be that in the closed system no inflow and outflow exist, but the only difference is the open or closed abdomen. This should be clarified.

·         equation (2): after the third “=”, the brackets are not correct

·         line 193: is (41° to 43°) correct?

·         line 257: brackets o.k.?

Author Response

Comments 2

Comments 1 :

Caused by the addition of new text in the second version of the article one problem is now enhanced that should be considered.

In the introduction and in other parts of the article the authors should clarify and improve the explanation of the effect of hyperthermia in tumor treatment.

It is possible to “destroy” the tumor cells by heat, but for this, much higher temperatures than 41°-43° are necessary and it is usually called “thermoablation”. This could be done by HIFU, laser and RFA.

Answer 1 :
In constructing the sentences in the introduction, I seem to have created a lot of confusion. I reviewed the paper again with reference to what you pointed out. Please see lines 74-89, 99-102 (red) for corrections.

á´‘ Line 74-89 (red):

In addition, HIFU therapy, laser therapy, radiofrequency therapy, and radiation therapy methods are used for targeted treatment of tumors and are often used in combination with chemotherapy to increase the effectiveness of treatment. Chemotherapy is ad-ministered intravenously, but various treatment systems that allow for direct exposure of anticancer drugs are currently under investigation and development.

Among them, hyperthermic intraventricular chemotherapy (HIPEC) surgery is a system that uses high-temperature anticancer drugs and injects them into the abdominal cavity, and it is a treatment method that directly exposes anticancer drugs in residual tumors after surgical operation. Hyperthermia using high-temperature anticancer drugs has been shown to be highly effective against cancer cells when administered at temperatures ranging from 41°C to 43°C Celsius, whereas normal tissues can be damaged by temperatures above 46°C Celsius [12]-[14]. This hyperthermia increases the sensitivity of cancer to chemotherapy by increasing the penetration of chemotherapy on the peritoneal surface and impairing DNA repair. In addition, hyperthermia has direct cytotoxic effects by inducing apoptosis, activating heat shock proteins that act as receptors for natural necrotic cells, inhibiting angiogenesis, and promoting protein denaturation [15].

á´‘ Line 99-102:

However, if a 41°C to 43°C medication is injected into the abdominal cavity while using a hyperthermia system in the operating room, the intraperitoneal temperature will be lowered to below 41°C, and the therapeutic effect of hyperthermia will not be enhanced [12].

The rest of the additional answers are:

Based on the points you pointed out, I reviewed the paper again. The laser was investigated as having a high temperature (in particular, up to 50℃, which is much higher than 43℃ for skin cancer) [L 1]-[L 5], as the reviewer suggested. Thermoablation was mostly analyzed with high temperatures of 80-90 °C [L6]. RFA was investigated with tissue damage temperatures above 46°C (60 min beam irradiation), 50-52°C (4-6 min beam irradiation), and 60-100°C [L7]. HIFU is usually over 50-55℃, and in special cases, it is analyzed by irradiating heat up to 80℃ [L8, L9]. In HIPEC surgery, 43℃ is introduced through the inflow, so the temperature in the abdominal cavity is 41-42℃ [L10].

Therefore, it was confirmed that the temperature ranges for laser, RFA and HIFU are different compared to HIPEC.

The temperature range seems to have caused a lot of confusion in the introduction because there are temperature ranges for lasers, RFAs and HIFUs. Therefore, the temperature range was deleted. So I tried to clear up any confusion about the contents of the introduction.

In particular, the temperature range of radiation therapy was also excluded. The problem of DNA chain breakage by irradiation does not appear to be related to hipec surgery. Therefore, to avoid confusion, the sentence about radiation therapy has been removed from the introduction. You seem to have caused a lot of confusion by constructing sentences in the introduction. Therefore, please see lines 74-89 (red) for corrections.

[L1] Ali, M.R.; Ali, H.R.; Rankin, C.R.; El-Sayed, M.A. Targeting heat shock protein 70 using gold nanorods enhances cancer cell apoptosis in low dose plasmonic photothermal therapy. Biomaterials 2016, 102, 1–8.

[L2] Song, A.S.; Najjar, A.M.; Diller, K.R. Thermally Induced Apoptosis, Necrosis, and Heat Shock Protein Expression in Three-Dimensional Culture. J. Biomech. Eng. 2014, 136, 071006.

[L3] Zhu, X.; Feng, W.; Chang, J.; Tan, Y.-W.; Li, J.; Chen, M.; Sun, Y.; Li, F. Temperature-feedback upconversion nanocomposite for accurate photothermal therapy at facile temperature. Nature Commun. 2016, 7, 10437.

[L4] Kim, M.; Kim, G.; Kim, D.; Yoo, J.; Kim, D. C.; Kim, H. Numerical study on effective conditions for the induction of apoptotic temperatures for various tumor aspect ratios using a single continuous-wave laser in photothermal therapy using gold nanorods. Cancers, 2019, 11, 764.

[L5] West Conor, L.; Austin C. V. Doughty, Liu, Kaili, Chen Wei, R. Monitoring tissue temperature during photothermal therapy for

Cancer. J. BioX Res. 2019, 2, 159–168.

[L6] Vihko Kimmo, K. Raitala, R. Taina, E. Endometrial thermoablation for treatment of menorrhagia: comparison of two methods in outpatient setting. Acta Obstet Gynecol Scand, 2003, 82, 269-274.

[L7] Voizard, N.; Cerny, M.; Assad, A.; Billiard, J. S.; Olivié, D.; Perreault, P.; Kielar, A.; Do Richard, K. G.; Yokoo, T.; Sirlin Claude, B.; Tang, A. Assessment of hepatocellular carcinoma treatment response with LI-RADS: a pictorial review. Insights into Imaging (2019) 10:121.

[L8] V. A. Khokhlova, O. V. Bessonova, M. V. Averiyanov, J. E. Soneson, and R. O. Cleveland, “Modeling of nonlinear shock wave propagation and thermal effects in high intensity focused ultrasound fields,” in Proc. 159th mtg. of Acoustical Soc. of America/ Noise-Con, Apr. 2010. Abstract: Journal of the Acoustical Society of America., vol. 123, p. 1827, 2010.

[L9] Karaböce, B. Investigation of thermal effect by focused ultrasound in cancer treatment. IEEE Instrumentation and Measurement Magazine. 2016, 19, 20-64.

[L10] Glockzin, G.; Schlitt Hans, J.; Piso, P. Peritoneal carcinomatosis: patients selection, perioperative complications and quality of life related to cytoreductive surgery and hyperthermic intraperitoneal chemotherapy. World Journal of Surgical Oncology, 2009, 7, 1-8.

Comments 2 :

In the temperature range of 40°-43° - called hyperthermia – the effect is not directly the cell destroying but the amplifying of the direct cytotoxic effect of chemotherapy and/or radiation therapy. This is also in HIPEC the effect and the goal. Therefore, formulations as “damage tumor cells by heat” or “thermal necrosis” are misleading in this temperature range and should be corrected.

Answer 2 :

Based on your good points, we've made the following changes. Please see lines 406-413 in the discussion. Thank you very much. Please see lines 413-420 (red) for corrections.

In this paper, a control system is proposed to maintain and compensate the treatment temperature in HIPEC surgery using high temperature drugs in ovarian cancer surgery.

The small tumor tissue remaining after ovarian cancer surgery is difficult to treat surgically.

 Therefore, the method proposed in this study can increase the therapeutic effect of hyperthermic chemotherapy by injecting and circulating drugs in the abdominal cavity and keeping the treatment temperature (41 degrees to 43 degrees) constant. Also, by preventing the temperature from rising above a certain temperature (46 degrees), it is possible to reduce side effects that may occur due to damage to normal tissues.

Some further remarks:

Comments 3 :

line 23: substances are

Answer 3 : It's corrected. Note the yellow in line 23. thank you so much.

Comments 4 :

line 35: I think: …  If Tdif is 44°C or higher  …

Answer 4 : Fixed (green on line 35). Thank you very much for pointing it out.

Comments 5 :

line 75: radiation therapy does not cause heat necrosis but DNA damage that results in necrosis

Answer 5 :

When I read it again, there was something wrong. So it seems that the reviewer pointed it out well. So, I deleted the content and edited it again. See lines 74-89 in the introduction. thank you so much.

Comments 6 :

line 102: “reducing the effect of the therapeutic agent …” instead of “ making it impossible …” and no thermal (see above)!

Answer 6 : Edited introduction, deleted sentence about comment. Thank you very much for your comments.

Comments 7 :

 lines 115-126: In the text the main difference between close and open HIPEC seems to be that in the closed system no inflow and outflow exist, but the only difference is the open or closed abdomen. This should be clarified.

Answer 7 :

Based on your good points, we've made the following changes. Please see lines 114-134.
HIPEC treatments are categorized into open and closed techniques, as shown in Figure 1. The closed technique involves inserting and securing the inflow and outflow catheters into the abdominal cavity before injecting chemotherapy, then closing the abdominal wall and injecting the hot chemotherapy solution to allow perfusion into the abdominal cavity. The main disadvantage of the closed technique is the uneven distribution of chemotherapeutic agents in the abdominal cavity, resulting in fluid retention and the ac-cumulation of toxic concentrations of drugs and heat [21].

In comparison, the HIPEC open method or "coliseum" technique involves injecting the hyperthermic chemotherapy solution into the abdominal cavity while the abdomen is open. The advantages of the open approach include the surgeon's direct access to the abdominal cavity with an inflow catheter during the administration of the hyperthermic agent, which allows for rapid and uniform control of the temperature and distribution of the drug in the abdominal cavity by manipulating fluids and bowel. Care also needs to be taken to ensure that all peritoneal surfaces are uniformly exposed during treatment and to avoid dangerous temperatures or excessive exposure to normal tissue. Potential disadvantages of this procedure include rapid loss of heat, requiring more effort to maintain ideal temperatures, and potential exposure of the surgeon and operating staff to chemo-therapeutic agents by direct contact and aerosolized particles [22].

For the use of these techniques, a HIPEC system consists of a fluid box, a heat ex-changer, and a filter, as shown in Figure 2. It is further composed of inlet and outlet catheters connected to the system [15], [21]-[23].

Comments 8 :

 equation (2): after the third “=”, the brackets are not correct

Answer 8 : It's corrected. thank you

Comments 9 :

line 193: is (41° to 43°) correct?

Answer 9 : I made a mistake. Thanks for pointing it out. It's corrected. See line 202 (green).

Comments 10 :

line 257: brackets o.k.?

Answer 10 : It's corrected. Line 266 (note green). Thanks for pointing it out.
